# Left Ventricular Non-Compaction: Review of the Current Diagnostic Challenges and Consequences in Athletes

**DOI:** 10.3390/medicina56120697

**Published:** 2020-12-14

**Authors:** Giuseppe Femia, Christopher Semsarian, Samantha B. Ross, David Celermajer, Rajesh Puranik

**Affiliations:** 1Sydney Medical School, Faculty of Medicine and Health, The University of Sydney, Sydney 2006, Australia; christopher.semsarian@sydney.edu.au (C.S.); s.barratt@centenary.org.au (S.B.R.); David.Celermajer@health.nsw.gov.au (D.C.); raj.puranik@cmrs.org.au (R.P.); 2Agnes Ginges Centre for Molecular Cardiology Centenary Institute, Camperdown 2050, Australia; 3Department of Cardiology, Royal Prince Alfred Hospital, Camperdown 2050, Australia

**Keywords:** left ventricular non-compaction, echocardiography, cardiac magnetic resonance

## Abstract

Left ventricular non-compaction (LVNC) is a complex clinical condition with no diagnostic gold standard. At present, there is trepidation about the accuracy of the diagnosis, the correlation to clinical outcomes and the long-term medical management. This article reviews the current imaging criteria, the limitations of echocardiography and cardiac magnetic resonance and the consequences of LV hypertrabeculation in athletes.

## 1. Introduction

Left ventricular non-compaction (LVNC) is an unclassified cardiomyopathy characterized by an abnormally thick trabeculated non-compacted myocardial layer with adjacent deep intra-trabecular recesses and a thin compacted myocardial layer [1]. Although the underlying etiology of LVNC remains uncertain, emerging evidence suggests that the excessive trabeculation may result from a disturbance in the compaction process during early myocardial development [2]. It is believed that pathological hypertrabeculation occurs from a failure of the final phase of this process [3]. During normal development, myocytes project radially towards the ventricular cavity from base to apex and septum to lateral wall, where they are covered by an endocardial layer [4,5]. Although specific genes have been found to contribute, there is pronounced genetic variability and a low diagnostic yield of genetic testing [6,7]. For some individuals, abnormal trabeculations may develop in conjunction with other cardiovascular or systemic conditions [8,9]. In some athletes, it remains unclear whether the abnormal myocardial morphology is representative of pathological LVNC or an epiphenomenon of cardiac adaptations from increased loading conditions [10]. Overall, there is heterogeneity in the clinical manifestations from no symptoms to ventricular arrhythmias, left ventricular (LV) dysfunction, stroke and/or cardiac death [11]. 

Heightened awareness and improved imaging techniques has led to inaccurate diagnosis, clinical challenges and unwarranted restriction from competitive sport [12,13]. At present, there are several 2-dimensional echocardiographic (2-d echo)- and cardiac magnetic resonance (CMR)-based criteria, but no diagnostic “gold standard” or specific clinical guidelines to help differentiate physiological hypertrabeculation from pathological LVNC [14,15,16,17,18,19,20,21]. 

This review highlights the strengths and weaknesses of the current diagnostic criteria, the future directions of evaluating LV hypertrabeculation and the implication for athletes.

## 2. Results

### 2.1. Echocardiographic Criteria

Due to its low cost and widespread availability, 2D-echo is usually the first investigation in the evaluation of LV hyper-trabeculation. Presently, there are four 2D-echo-based criteria that are commonly used, but none are considered as the gold standard (Table 1). The first criterion was developed by Chin et al and defines LVNC as an epicardial compacted myocardium layer (Χ) to endocardial non-compacted layer ratio (Y) ≤0.5 in the end-diastole [14]. Although this criterion has been shown to have the greatest sensitivity and specificity amongst the four 2D-echo criteria, it is not widely used in clinical practice [22]. The next criterion was developed by Jenni et al and defines LVNC as a non-compacted to compacted myocardial ratio >2.0 on short-axis images taken in end-systole [15]. In addition, the criterion requires a colour Doppler flow through prominent trabeculations in communication with intertrabecular space in the absence of co-existing cardiac abnormality. When using this criterion, it should be noted that this criterion had the lowest reproducibility and diagnostic validity amongst all four criteria [23]. A review of these two criteria suggested that the myocardial thickness is best measured in end-diastole on short-axis images and a non-compacted to compacted ratio >2.0 is diagnostic of LVNC in accord with CMR measurements [24]. The third criteria defined LVNC by the presence of three or more trabeculations along the LV endocardial borders (different from papillary muscle, false tendons and aberrant muscle bands which move in synchrony with compacted myocardium) [16]. This criterion may lead to overdiagnosis as it was largely extrapolated from a large post-mortem cohort [25]. Finally, a recent study by Gerbhard et al compared myocardial thickness in patients with LVNC and moderate to severe aortic valve stenosis, and found that a compacted myocardial thickness of less than 8 mm differentiated pathological LVNC from physiological hyper-trabeculation [17].

### 2.2. Limitations of Echocadiography

Despite the widespread use of 2D-echo in the evaluation of hyper-trabeculation, the current criteria have several limitations. First, 2D-echo has limited spatial resolution and a poor ability to image the entire left ventricular cavity, making it difficult to differentiate non-compacted from compacted myocardium. For example, the LV apex, which is a commonly affected segmented can be difficult to visualize and differentiate from benign LV trabeculation [26]. The limited spatial resolution of 2D-echo has been shown to result in poor diagnostic accuracy, particularly when in the evaluation of certain groups, such as athletes and pregnant women. In one study, the authors scanned more than 1000 asymptomatic athletes and found that 18% had increased LV trabeculation and 8% fulfilled the criteria for LVNC [10]. In a second study, the same authors assessed 102 women with normal echocardiograms and found the 25% developed de novo trabeculation during pregnancy [8]. Interestingly, at 24 months post-partum, 73% had complete resolution of trabeculations. These criteria were developed using small patient cohorts and derived from non-prospective studies. Although some of the criteria have been validated in subsequent studies, there is ongoing apprehension about their accuracy and diagnostic consensus. One recent study assessed the accuracy of three 2D-echo criteria for the diagnosis of non-compacted myocardium in patients with heart failure compared to patients without heart disease and found variability in the diagnostic prevalence of the three diagnostic criteria [27]. The Chin criteria had the highest rate of diagnosis (79% vs, 64% vs. 53%, respectively). Of most concern was that the correlation between the criteria was weak, as only 30% of patients satisfied all three criteria. Most importantly, there remains a limited correlation between diagnostic 2D-echo criteria and clinical outcomes. The criteria define pathological LVNC only by abnormal morphological appearance and do not include other ventricular parameters such as LV systolic function or volumes. Several studies have shown that 2D-echo has poor correlation with clinical outcomes, as adverse events do not appear to be influenced by the degree of LV trabeculation detected by 2D-echo but by the presence of LV systolic dysfunction and LV scar [28]. 

### 2.3. Cardiac Magnetic Resonance Criteria

CMR is generally used to supplement and confirm 2D-echo findings by providing better spatial resolution in all LV segments, detailed visualization of cardiac morphology, robust volumetrics and the ability to identify fibrosis with late gadolinium enhancement (LGE). There are currently four validated CMR-based criteria but, again, no gold-standard has been established; Table 2. The first criterion uses end-diastolic images in any long-axis LV view and defines LVNC as a non-compacted to compacted myocardial ratio >2.3 [18]. This criterion has been shown to have a high prevalence rate and poor correlation to clinical outcomes [29]. Moving away from morphological assessment, the criteria proposed by Jacquier and colleagues measured LV contours and assessed LV volume, LV ejection fraction, LV mass and LV trabeculation in patients with LVNC, hypertrophic cardiomyopathy and dilated cardiomyopathy [19]. The authors found that a trabecular mass ≥20% of total LV mass was sensitive and specific to the diagnosis of LVNC. One criticism of this criteria was the inconsistency of including papillary muscle mass into the trabeculated mass and, as a result, significantly increasing the threshold to diagnose significant LVNC. Subsequently, Stacey and colleagues compared the Jacquier criteria to other definitions and suggested increasing the diagnostic threshold for LVNC from 20% to 40% [30]. The authors also found that end-systolic measurements had stronger associations with clinical events such as systolic dysfunction, congestive heart failure and death compared to measurement made in end-diastole. Adding to these data, another group also measured LV contours but added new qualitative and semi-quantitative parameters such as LGE [20]. The authors found four diagnostic parameters with strong positive predictive values for diagnosing and discriminating LVNC from other cardiomyopathies; percentage of non-compacted mass > 25%, total indexed non-compacted myocardial mass > 15 g/m^2^, a non-compacted to compacted myocardial ratio ≥3:1 in segments 1–3 or 7–16 excluding the apex and a non-compacted to compacted myocardial ration ≥2:1 in segments 4–6. The final criterion was proposed by Captur and colleagues using a method of quantifying complex geometric patterns in biological structures called fractal analysis [21]. The authors found that a fractal dimension (a unitless measure index of how completely the object fills space) ≥1.3 was consistent with LVNC. Although fractal analysis is not regularly used in clinical practice, recent studies have shown that high fractal dimension was observed in all patients diagnosed with LVNC [31] and can be used as a biomarker to distinguish LVNC from dilated cardiomyopathy [32]. 

### 2.4. Limitations of Cardiac Magnetic Resonance

Several concerns have been highlighted about CMR and the current criteria. First, the four criteria are based on small cohorts and the data are not prospectively derived. Even though the criteria have been shown to accurately differentiate LVNC from other cardiomyopathies, none of the criteria have been correlated with clinical outcomes. In fact, only one study showed a strong association between end-systolic measures of LVNC and adverse events such as congestive heart failure [30]. In addition, none of the current CMR-based criteria include other LV parameters such as LV ejection fraction or LV scar in the assessment of LVNC, despite recent evidence demonstrating that LV non-compaction alone is not predictive of clinical events. In a study of 113 patients with a diagnosis of LVNC, the degree of LV trabeculation does not have prognostic impact over and above LV dilation, LV systolic dysfunction and the presence of LGE [12]. In another study, the authors evaluated hyper-trabeculation in 162 consecutive patients with dilated cardiomyopathy and found that only LV ejection fraction and scar, as determined by LGE, were independent predictors of MACE-free survival [26]. Although CMR can more easily differentiate compacted from non-compacted myocardium throughout the entire LV cavity, the rate of diagnosis has been shown to be higher compared to 2D-echo. A recent systematic review of fifty-nine studies reporting LVNC prevalence in adults found a higher prevalence with CMR imaging and criteria [13]. Given the poor correlation with clinical outcomes, there are concerns that not all hyper-trabeculation is pathological. One study by Kawel et al applied the Petersen criteria to a large group of healthy volunteers without cardiac disease and found that a non-compacted to compacted ratio >2.3 was common and not necessarily indicative of pathological disease [33]. In fact, even when the authors raised the ratio of non-compacted to compacted myocardium to >3, there was no correlation with clinical outcomes. Utilizing CMR as the primary imaging modality remains difficult due to the limited availability in non-urban areas, high running costs and relatively longer acquisition times. 

### 2.5. Hypertrabeculation and Athletes

In the evaluation of athletes, echocardiographic studies have demonstrated a high prevalence of LV hypertrabeculation, fulfilling at least one of the diagnostic criteria. The authors of the PESA (Progression of Early Subclinical Atherosclerosis) study assessed the relationship between LVNC phenotype on CMR imaging and accelerometer-measured physical activity and found that the prevalence of an LVNC phenotype according to several CMR criteria was significantly higher in those with the highest physical activity quintile [34]. The association between physical activity and LVNC phenotype was independent of LV volume. As a consequence, distinguishing pathological LVNC from physiological remodeling remains a diagnostic challenge. In a cross-sectional echocardiographic study, a group of more than 1100 athletes were found to have a higher prevalence of LV hypertrabeculation compared to controls (18.3% vs. 7.0%) but during a long-term follow-up, all athletes were asymptomatic and free of adverse events [10]. In a subsequent study of more than 2500 athletes, 36 were found to have prominent trabeculations that satisfied at least one echocardiographic criterion. Of these, only three patients were considered to be pathological, with either LV dysfunction, a family history of LVNC, or a known pathogenic gene mutation [35]. Finally, in other studies, there were no reported cases of sudden cardiac death in athletes with hypertrabeculation [36,37].

## 3. Management

Managing LVNC presents a significant clinical challenge given the variability in manifestations and the limited long-term efficacy of specific treatments. Although most patients with LVNC remain asymptomatic, it is important to review patients regularly with cardiac imaging, as some may be at risk of heart failure, stroke and/or sudden cardiac death. In particular, those with reduced LV function should be reviewed frequently and treated with evidence-based, guideline-directed pharmacologic therapy. As per guidelines, an intracardiac defibrillator should be offered to those who survive an episode of sustained ventricular tachycardia (VT) or sudden cardiac arrest [38]. Successful cardiac transplantation has been reported in some patients and should be considered for those with end-stage heart failure [39]. Even though the event rate of stroke is 1–2% per year, the optimal medical strategy in those who do not meet the standard criteria for anticoagulation remains uncertain given the scarcity of data [40]. However, patients with a prior cardioembolic event, evidence of an intracardiac thrombus and/or documented atrial fibrillation should be treated with anticoagulation consistent with standard recommendations for cardiogenic embolism [24]. For athletes, it has been suggested that only those that meet the LVNC criteria with impaired left ventricular function should be prohibited from participating in sport, while asymptomatic athletes with normal ventricular function do not require restrictions on activity [41,42].

## 4. Discussion

In the evaluation of hyper-trabeculation, the diagnosis of LVNC is limited by the lack of standardized imaging criteria and poor correlation with clinical outcomes, leaving clinicians to rely on small single-center studies. We know from previous data that criteria based on morphological parameters alone are inadequate to prognosticate patients. As a result, assessments have shifted away from static single-plane, two-dimensional measurements towards the quantification of trabeculated volume or mass [19,20]. The addition of cardiac volumes, LV systolic function and the presence or absence of LV scar to pre-existing patient factors (family history of LVNC or sudden cardiac death, symptomatic heart failure or documented VT) may help stratify individuals at higher risk of complications, enhance clinical management and improve long-term outcomes. Recently, studies examining patients with isolated LVNC have found that only those with impaired LV systolic function and/or LV scar are at high risk of adverse events [43]. Moving forward, further information from large clinical registries and prospective data on ethnic variation and the dynamic changes seen in athletes and during pregnancy is required.

In an effort to improve the overall diagnostic accuracy, a clinical algorithm has been developed to help guide clinicians in the assessment of patients with suspected LVNC [44]. The authors have proposed that LVNC can be evaluated with either 2-d echo or CMR imaging and diagnosed with any criteria. Following this suggestion and based on the most current evidence, we that recommend all patients with hyper-trabeculation fulfilling any of the imaging-based criteria either 2D echo or CMR should be assessed for impaired LV function and LGE. For these patients, clinical management should continue as per standard clinical guidelines and should include family screening and/or genetic testing. For those with normal LV function and no LGE, patients should be screened for a family history of LVNC or SCD, syncope, ventricular arrhythmias and thromboembolic events to help predict the risk of adverse events and the need for further assessment. In the absence of these risk factors, patients can be reassured with less intensive long-term follow up; Figure 1.

## 5. Conclusions

LVNC is a heterogenous condition with no universally accepted diagnostic criteria or gold standard. In the evaluation of patients and athletes with LVNC, it is important to take a comprehensive history, rely on more than one diagnostic method and include LV parameters such as LV function and LGE. In those with clinical suspicion of LVNC, echocardiography remains the first imaging modality; however, once hypertrabeculation has been identified, CMR should be used to evaluate LV ejection fraction and assess for LGE. For those with impaired LV function and/or LGE, prohibiting vigorous sports activities should be considered and management goals should be based on clinical symptoms such as ventricular arrhythmias, syncope or thromboembolic events. 

## Figures and Tables

**Figure 1 medicina-56-00697-f001:**
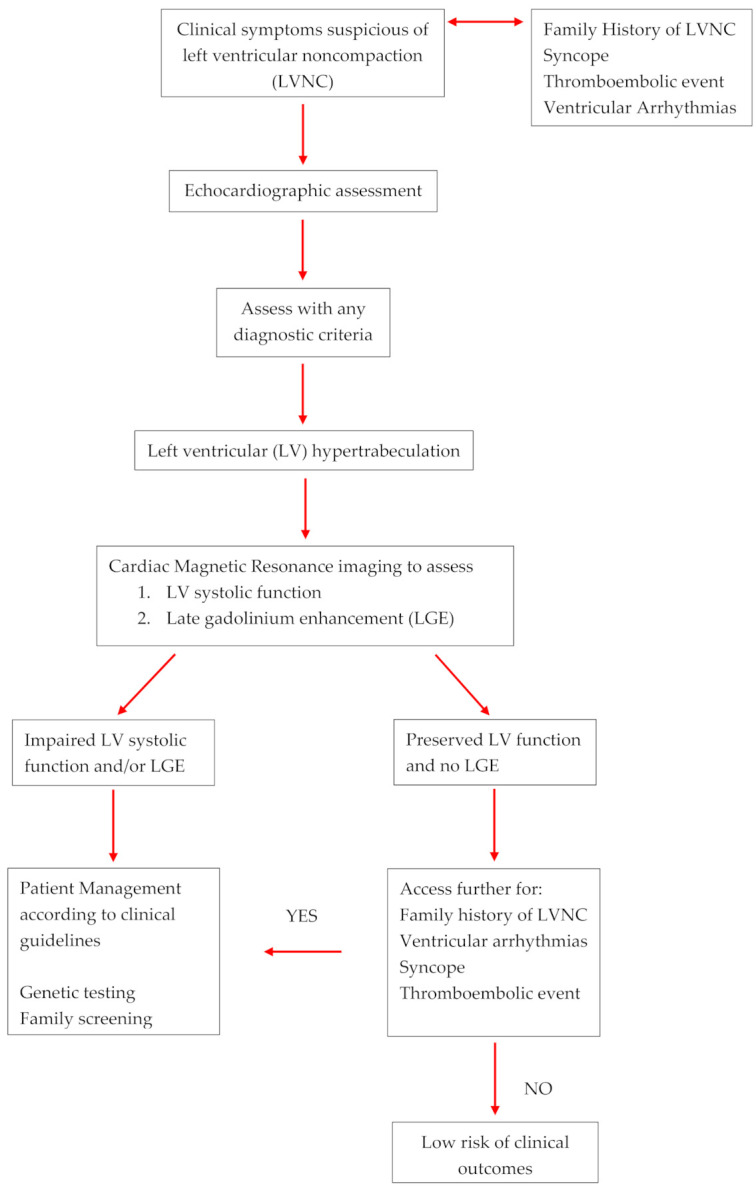
Diagnostic Algorithm for left ventricular hyper-trabeculation.

**Table 1 medicina-56-00697-t001:** Echocardiographic-based Left Ventricular Non-Compaction (LVNC) diagnostic criteria.

	Chin	Jenni	Stollberger	Gebhard
**Year**	1990	2001	2002	2012
**Total Patients;** **Patients with LVNC**	8; 8	34; 34	62; 62	123; 41
**Selection Criteria**	Patients referred for echo and satisfied criteria	Patients referred for transthoracic echo	Patients referred for echo demonstrating >3 trabeculations distal to papillary muscle in the four-chamber view	Patients with LVNC by other echo criteria, severe aortic stenosis and matched controls
**Age Range**	11 months to 22.5 years	16 to 75 years	18 to 75 years	20 to 52 years
**Description of Criteria**	1. 2-layered structure with an epicardial compacted (C) and endocardial noncompacted (NC) layer ≤0.5	1. 2-layered structure 2. Noncompacted endocardial layer Colour Doppler evidence of inter-trabecular recesses supplied by intraventricular blood 3. NC/C ≥2 4. No coexisting cardiac abnormalities	1. 2-layered structure 2. >3 trabeculations protruding from LV wall apically to papillary muscle in 1 imaging plane 3. NC/C ≥2	1. 2-layered myocardium 2. Maximal systolic compacted thickness <8 mm
**View**	Parasternal Short-axis view	Parasternal Short-axis view	Non-standard views	Parasternal Short-axis view
**Phase**	End-systole	End-diastole	End-diastole	End-systole
**Correlation with Clinical**	No	No	No	No
**Strengths**	Widely available Cost-effective Short scanning time	**Weaknesses**	Participants had wide age range Results based on small cohorts Studies not prospectively derived Image quality depedant on body habitus Oversensitive in certain populations Non-specific in low-risk populations	

**Table 2 medicina-56-00697-t002:** Cardiac magnetic resonance-based left ventricular non-compaction diagnostic criteria.

	Petersen	Jacquier		Grothoff		Captur
**Year**	2005	2010		2012		2013
**Total Patients; Patients with LVNC**	177; 7	64; 16		57; 12		135; 30
**Selection Criteria**	Patients with either transthoracic echo or cardiac magnetic resonance (CMR) documentation of 2-layered myocardium; athletes, patients with hypertrophic or dilated cardiomyopathy and matched controls	Patients fulling Jenni et al. criteria for LVNC; patients with hypertrophic or dilated cardiomyopathy and Controls		Patients with echo LVNC (Jenni criteria) plus one of the following: LVNC in first-degree relatives, neuromuscular disorder or complications such as systematic embolization or regional wall motion abnormalities or ventricular abnormalities; patients with hypertrophic or dilated cardiomyopathy and matched controls		Patients with echo evidence of LVNC (Jenni criteria) plus one of the following: positive family history, associated neuromuscular disorder, regional wall motion abnormality, LVNC related complications including matched controls
**Age Range**	14 to 25 years	25 to 74 years		11 to 71 years		18 to 85 years
**Description of Criteria**	1. 2-layered myocardium with a compacted epicardial and noncompacted endocardial layer 2. NC/C ratio ≥ 2.3 in any long-axis LV image	1. Total left ventricular (LV) trabeculated mass ≥20% of the global LV mass		1. Percentage of non-compacted mass > 25% 2. Total indexed myocardial mass > 15 g/m^2^ 3. A non-compacted to compacted myocardial ratio ≥ 3:1 in segments 1–3 or 7–16 excluding the apex 4. A non-compacted to compacted myocardial ration ≥ 2:1 in segments 4–6		1. Fractal analysis with elevated fractal dimension-global LV trabecular complexity as a continuous variable
**View**	Any long-axis image	Short-axis stack		Short-axis stack		Short-axis stack
**Phase**	End-diastole	End-diastole		End-diastole		End-diastole
**Outcomes**	No	No		No		No
**Strengths**	Superior signal to noise ratio Unlimited imaging planes Superior tissue characterization High sensitivity and specificity		**Weaknesses**		Not widely available Requires expertise Expensive Results based on small cohorts Studies not prospectively derived Over sensitive in certain populations	

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
