# Peer review of "Left Ventricular Non-Compaction: Review of the Current Diagnostic Challenges and Consequences in Athletes"

_medicina, 2020, doi:10.3390/medicina56120697_

Round 1
Reviewer 1 Report
Dear Authors,
I would first congratulate you for your work. The medical problem you present in your article is important from clinical point of view. Left Ventricular Non-Compaction is still an object of large and frequently arguable discussions in terms of the diagnostic criteria and the optimal therapeutic approach.
I suggest that you review it for the following minor corrections:
1. Line 29: “…..may develop be present…..” – perhaps, you should add “and” between “develop” - “be present”
2. Please, spell “LVNC” and “LV” not only in the abstract, but also the first time you use them in the main text. The same for CMR
Author Response
Point 1:
“…..may develop be present…..” – perhaps, you should add “and” between “develop” - “be present”
Response 1 (highlighted yellow in manuscript)
Lines 79 - 80
For some individuals, abnormal trabeculations may develop in conjunction with other cardiovascular or systemic conditions.
Point 2: Please, spell “LVNC” and “LV” not only in the abstract, but also the first time you use them in the main text. The same for CMR
Response 2: (highlighted yellow in manuscript)
Line 69 Left ventricular non-compaction (LVNC)
Line 84 left ventricular (LV)
Line 88 - 89 cardiac magnetic resonance (CMR)

Reviewer 2 Report
I have read a current review paper by Femia et al with much interest. LVNC is not so rare, obviously, and may inflict serious conseqences if not diagnosed on time and left without treatment. I would really appreciate a paragraph regarding treatment of LVNC in the manuscript. That would give a full insight in LVNC approach, also in athletes. The article is definitely worth to be published and I hope the authors would add a short paragraph about treatment pathways.
Author Response
Comment 1:
I have read a current review paper by Femia et al with much interest. LVNC is not so rare, obviously, and may inflict serious conseqences if not diagnosed on time and left without treatment. I would really appreciate a paragraph regarding treatment of LVNC in the manuscript. That would give a full insight in LVNC approach, also in athletes. The article is definitely worth to be published and I hope the authors would add a short paragraph about treatment pathways.
Response 1: (highlighted in yellow in manuscript)
Please see the attachment
Lines 236 - 252
Managing LVNC presents a significant challenge given the variability in clinical manifestations and the limited long-term efficacy of specific treatments. Although most patients with LVNC remain asymptomatic, it is important to review patients regularly with cardiac imaging as some may be at risk of heart failure, stroke and/or sudden cardiac death. In particular, those with reduced LV function should be treated with evidence-based, guideline directed pharmacologic therapy. An intracardiac defibrillator should be offered to those who survive an episode of sustained ventricular tachycardia (VT) or sudden cardiac arrest (39). Successful cardiac transplantation has been reported in some patients and should be considered with end-stage heart failure (40). Even though the event rate of stroke is 1-2% per year, the optimal medical strategy in those who do not meet standard criteria for anticoagulation remains uncertain given the scarcity of data (41). However, patients with a prior cardioembolic event, evidence of an intracardiac thrombus and/or documented atrial fibrillation should be treated with anticoagulation consistent with standard recommendations for cardiogenic embolism (24). For athletes, it has been suggested that only those that meet LVNC criteria with impaired left ventricular function should be prohibited from participating in sport while asymptomatic athletes with normal ventricular function do not require restrictions on activity (42, 43).

Reviewer 3 Report
Fermia et al have made a well documented and well written review of Left Ventricular Non-Compaction, that, in my opinion, deserves to be published.
No major concerns were found.
Author Response
No revisions suggested